# Vascular Repair by Grafting Based on Magnetic Nanoparticles

**DOI:** 10.3390/pharmaceutics14071433

**Published:** 2022-07-08

**Authors:** Xin Liu, Nan Wang, Xiyu Liu, Rongrong Deng, Ran Kang, Lin Xie

**Affiliations:** Third School of Clinical Medicine, Nanjing University of Chinese Medicine, Nanjing 210028, China; liuxin@njucm.edu.cn (X.L.); wangnan082020@163.com (N.W.); liuxiyu1992@live.com (X.L.); rongrdeng@126.com (R.D.)

**Keywords:** magnetic nanoparticles, vascular repair, magnetic responsiveness, non-invasive way

## Abstract

Magnetic nanoparticles (MNPs) have attracted much attention in the past few decades because of their unique magnetic responsiveness. Especially in the diagnosis and treatment of diseases, they are mostly involved in non-invasive ways and have achieved good results. The magnetic responsiveness of MNPs is strictly controlled by the size, crystallinity, uniformity, and surface properties of the synthesized particles. In this review, we summarized the classification of MNPs and their application in vascular repair. MNPs mainly use their unique magnetic properties to participate in vascular repair, including magnetic stimulation, magnetic drive, magnetic resonance imaging, magnetic hyperthermia, magnetic assembly scaffolds, and magnetic targeted drug delivery, which can significantly affect scaffold performance, cell behavior, factor secretion, drug release, etc. Although there are still challenges in the large-scale clinical application of MNPs, its good non-invasive way to participate in vascular repair and the establishment of a continuous detection process is still the future development direction.

## 1. Introduction

Vascular damage caused by external force invasion or vascular disease is the leading cause of death worldwide [1,2,3]. For minor vascular damage clinically, simple surgical sutures and medical treatment can be used for treatment. However, severe trauma or insufficient blood supply caused by large-scale defects can cause serious consequences, such as claudication, sores, organ dysfunction, necrosis, or even death [4,5,6]. At present, vascular grafts are widely used clinically to prevent severe vascular injury, such as polyurethane [7], polyester [8], expanded polytetrafluoroethylene (ePTFE) [9], etc. However, the repair of vascular structures is not a single graft but a complex repair process. Vascular repair mainly includes three stages: inflammation, neointimal hyperplasia, and vascular remodeling, in which dysfunction at any stage will affect the vascular repair [10,11,12]. Due to their unique magnetic responsiveness [13,14], magnetic nanoparticles (MNPs) are considered an effective non-invasive technical means to assist in various stages of vascular repair and have achieved good results in both diagnosis and treatment [15,16,17]. For example, MNPs are used as magnetic resonance contrast agents to evaluate and monitor the fate and function of grafts with non-invasive imaging methods [18,19,20]; MNPs-labeled vascular endothelial cells (VECs) can be driven to seed at designated sites to accelerate endothelialization [16]; MNPs are used as a targeted anticoagulant drug carrier reduces thrombus formation [21,22], etc.

The application of MNPs in vascular repair mainly depends on their biological safety and magnetic response characteristics [23,24]. The currently developed MNPs contain various metals and metal oxides, but only a few MNPs are used in clinical research, mainly due to potential toxicity and rapid oxidation [25,26]. Iron oxide nanoparticles (IONPs) are commonly used MNPs [27,28,29,30], especially pure iron oxides such as magnetite (Fe_3_O_4_) and maghemite (γ-Fe_2_O_3_) [31,32]. Although IONPs doped with magnetically susceptible elements (e.g., MFe_2_O_4_ where M = Co, Mn, Ni, or Zn) and metal alloy nanoparticles (e.g., FeCo and FePt) revealed better magnetic responsiveness than pure IONPs, their potential toxicity greatly limits their applications [33,34]. Therefore, MNPs involved in vascular repair generally refer to the core of pure nano-iron oxide crystals and the surface coating with high biological safety to avoid reactive oxygen species (ROS) [27,35,36].

At present, the latest literature search found that the application of MNPs in vascular repair mainly focuses on a single aspect, such as magnetic resonance imaging (MRI) characteristics, magnetically driven targeted drug delivery, magnetic hyperthermia, etc., which often makes readers unable to fully understand the role of MNPs in all stages and causes of vascular repair. In fact, as a scaffold for large-segment vascular defects, the use of MNPs to improve the performance of scaffolds has rarely been discussed. Therefore, it is necessary to supplement this topic to give readers a more comprehensive understanding of the various roles of MNPs in vascular repair; for example, MNPs can improve the performance of vascular scaffolds, including mechanics, porosity, degradation, etc., but there is no focused review to discuss in-depth, and this review can inspire them to focus on this aspect.

In this manuscript, we briefly introduce the classification and magnetic response characteristics of MNPs currently obtained in the research and overview their most prominent (pre-)clinical applications in vascular repair. Regarding the application of MNPs in vascular repair, currently, it is mainly used as individual MNPs or MNPs-labeled assemblies (e.g., MNPs labeled cells, MNPs assembled scaffolds, MNPs-loaded drug delivery, etc.). Finally, we also discussed the future development trend of MNPs in vascular repair and practical strategies to expand the scope of application.

## 2. Overview of Magnetic Nanoparticles (MNPs)

### 2.1. Classification and Composition of MNPs

Magnetic nanoparticles (MNPs) are a kind of intelligent nano-magnetic material [29,34]. They have the unique properties of nanomaterials, such as small particle size [37], large specific surface area [38], high coupling capacity [39], magnetic responsiveness [40], and superparamagnetism [41], which can make them aggregate and locate in a constant magnetic field and absorb electromagnetic waves to generate heat in an alternating magnetic field. Using these properties, MNPs are used in biomarking and separation, magnetic resonance imaging (MRI), tissue repair, drug carriers, and disease diagnosis and treatment [27,42,43]. MNPs designed for biomedical applications are mainly divided into single-component metal nanoparticles (e.g., Fe, Ni, Co, Mn, and Zn) (Figure 1A), metal alloy nanoparticles (e.g., FeCo and FePt) (Figure 1B), metal oxide nanoparticles (e.g., Fe_3_O_4_, γ-Fe_2_O_3_, CoFe_2_O_4_, and MnFe_2_O_4_) (Figure 1C), and heterostructures (e.g., Fe_3_O_4_@Au, γ-Fe_2_O_3_@PSC, γ-Fe_2_O_3_@SiO_2_) [44] (Figure 1D). However, only a few of these types of nanoparticles have entered (pre-) clinical studies, mainly due to their potential toxicity. Pure iron oxide is the most widely used nanomaterial in clinical applications due to its better biocompatibility and biological tolerance. However, the possible release of iron ions can produce ROS, and most of them will be coated with functional coatings (e.g., amino acids [45], polymers [46], fatty acids [47], oxides [46], metals [48], and multifunctional materials [49]) in scientific research and clinical applications [27]. Among these materials, polymer is the most commonly used coating material. In recent years, some natural and synthetic polymers, such as dextran [50], gelatin [51], alginate [52], chitosan [53], starch [54], albumin [55], casein [56], poly(ethylene glycol) (PEG) [57], poly(vinylpyrrolidone) (PVP) [58], poly(vinyl alcohol) (PVA) [59], polydopamine [60], poly(lactic-glycolic acid) (PLGA) [61], and dendrimers [62] have been used to coat MNPs in different forms. It is worth noting that PEG and dextran are the most widely used coatings, mainly because they have good biosafety and will not be quickly recognized and cleared by macrophages in the liver and spleen during intravenous administration [63,64]. PEG and PEGylated coatings (such as PEGylated starch) have high biocompatibility and are usually used to prolong the intravascular circulation of MNPs. Two critical parameters in PEG coating, molecular weight and surface density, significantly affect the dispersity, stability, cytotoxicity, and blood circulation time of MNPs [54,65]. Likewise, dextran (C_6_H_10_O_5_) n (a coating agent under the trade names ferumoxtran and ferumoxides) and its derivatives carboxydextran (a coating under the trade name ferucarboran) and carboxymethyl dextran (a coating under the trade name ferumoxytol) agent) are also known as highly biocompatible agents. However, it must be remembered that while glucan is not directly toxic to cells, its degradation products may affect specific cellular processes [66,67,68]. At present, many coating-modified IONPs have been evaluated in (pre-) clinical trials, and some of them have entered the market (Figure 2A). However, some approved iron oxide nanoparticles were later withdrawn because of their serious side effects [31]. In order to increase the stability and multifunctional therapeutic effects of nanoparticles (such as including disease treatment and imaging), polymer micelle-based iron oxide nanosystems have also been developed, mainly including: (1) A shell composed of a hydrophilic shell; (2) magnetic nanocrystallites and hydrophobic cores of polymer hydrophobic segments; (3) drug/gene payloads for therapy; (4) ligands which target the shell surface [69] (Figure 2B).

### 2.2. Main Preparation Methods of MNPs

MNPs used in scientific research and clinical use usually need to be customized and regulated by specific methods according to different use targets to meet the diagnosis and treatment of target diseases. For example, MNPs suitable for tissue MRI should be superparamagnetic and have a particle size distribution of usually 10–200 nm because particles larger than 4 μm risk clogging the lungs, while particles smaller than 7 nm tend to leak out of vascular structures and be blocked by kidney clearance and excretion [70,71]; MNPs suitable for magnetic hyperthermia applications should have stronger magnetic responsiveness [72]; MNPs as drug delivery carriers require smaller particle size to pass through the body barrier [73]; MNPs as iron supplements or cell tracers require better stability and surface functionality to avoid oxidative stress caused by iron release and to facilitate labeling by cell recognition [74]. Therefore, to better adapt to the needs of target diseases and ensure the shape, size, crystallinity, and surface properties of MNPs, their preparation methods mainly include chemical, physical and biosynthetic methods [75,76,77]. However, due to the lack of ability of physical methods (including powder ball milling, electron beam lithography, aerosol, and vapor deposition) to control particle size in the nanometer range and the low yield and broad size distribution of biological methods, MNPs accounted for less than 10% of the synthetic method [75]. The chemical synthesis method is currently the most widely used preparation method because it can precisely control the size, composition, and structure of the obtained MNPs, including co-precipitation, high-temperature pyrolysis, microemulsion, sol-gel, and the technology that can be applied in industrialization is only for chemical co-precipitation [29]. The chemical co-precipitation technique is mainly based on finalizing customized MNPs by adding weak or strong bases by adjusting synthesis parameters (such as Fe^2+^/Fe^3+^ ratio, temperature, pH, type of salt, and type of alkaline) [66,78]. In this way, the prepared MNPs will achieve the following goals: (a) form a protective layer with good stability; (b) reduce the surface energy of nanoparticles to avoid agglomeration; (c) can be adjusted by nucleation and growth process to control the size of MNPs; (d) promote the directional growth of the core structure to form shaped MNPs.

## 3. MNPs for Vascular Repair

### 3.1. MNPs in Vascular Grafts

Vascular grafts (also called vascular scaffolds) have always been a clinically effective treatment strategy for large vascular defects caused by vascular disease or violent invasion [4,79,80], such as polyurethane (PU) [81], polyester (PET) [82], ePTFE [83], etc. Although these vascular grafts, when used in larger-diameter vessels (>6 mm) show satisfactory long-term performance, they display inferior performance in small-diameter vessels (<6 mm), mainly because they are prone to intimal hyperplasia (IH) and thrombogenesis [84]. Given the above-mentioned unsatisfactory factors and the shortcomings of secondary operations caused by non-degradability, it has always been a goal to seek natural materials with better biocompatibility to construct vascular grafts. Unfortunately, the many properties of natural materials cannot meet the application requirements of vascular grafts, such as mechanical strength [85], elasticity [16], and degradation [86,87]. MNPs are believed to effectively improve the performance of vascular grafts and provide more applications in other areas (such as MRI [43] or nano-modification [88]). Ghorbani et al. [89] synthesized INOPs by co-precipitation technique and they were evenly distributed in PLGA-gelatin scaffolds. The results showed that the added MNPs had no special effect on the pore morphology but slightly reduced the pore size distribution. The MNPs containing the construct had enhanced mechanical strength, but the absorption capacity and biodegradation rate were reduced. Our previous study [16] also proved that MNPs were evenly distributed in silk fibroin scaffolds by infiltration. The results showed that the obtained magnetic silk fibroin scaffolds significantly delayed the degradation rate and enhanced mechanical strength (Figure 3A). Lekakou et al. [90] found that gelatin/elastin gels are nanocomposite scaffolds with flattened elastin nanodomains embedded in a gelatin matrix that mimic the structure of the arterial media. They studied gelatin/“hydroxyapatite” (HA) nanocomposite scaffolds, and “HA” was generated in situ in solution. When a magnetic field of 9.4 T was applied, the HA particles and gelatin microfibrils in the gelatin were oriented perpendicular to the direction of the magnetic field, which provided a basis for the preparation of arterial vascular layered scaffolds. Mertens et al. [20] prepared three ultra-small superparamagnetic iron oxide nanoparticles (USPIO), which were subsequently directly colonized in collagen scaffolds by chemical cross-linking and used indirectly as imaging graft scaffolds. Imaging can also be performed in the case of acellular implants to visualize the degradation of collagen scaffolds in vivo, which is beneficial for analyzing the in vivo degradation cycle and mechanism of rapidly degrading natural materials. Currently, MNP-added vascular grafts are mainly used to improve mechanical strength, mainly based on the high modulus, abundant functional groups. and uniform dispersion of MNPs (Figure 3B) [91].

### 3.2. MNPs Regulate Vascular-Related Cell Behavior and Factor Expression

Vascular injury repair is a highly organized engineering that mainly involves three stages, including inflammation, neointima, and remodeling [10]. For the initial acute inflammation stage, many macrophages will migrate to the injury site and secrete various inflammatory factors (such as TNF-α, IL-6, MCP-1) to clear the damaged cell debris and play a defensive role. When entering the late stage of inflammation and transitioning to the neointimal stage (active re-endothelialization stage), macrophages secrete various repair cytokines (such as bFGF, VEGF, and TGF-β) to regulate the microenvironment at the injury site. Thus, it regulates the behavior of various cells involved in re-endothelialization, including adhesion, migration, proliferation, phenotype, and homing [92,93]. However, regardless of the stage, various related cells and factors are involved, and favorable cell behavior and factor secretion can rapidly remodel blood vessels. Numerous studies have proved that the unique magnetic properties of MNPs can regulate cell behavior and factor secretion, thereby promoting vascular remodeling [94]. MNPs regulate cell behavior and factor secretion in the following ways: (1) MNPs through the stimulation of labeled cells (Figure 4A); (2) MNPs-labeled cells respond to magnetic fields (Figure 4B); (3) MNPs bind to materials to affect adherent cell behavior and factors (Figure 4C); (4) MNPs indirectly affect the behavior and factor secretion of target cells by affecting related pathways (Figure 4D). Lshii et al. [95] assembled a magnetic cell sheet by combining Fe_3_O_4_ nanoparticles with mesenchymal stem cells (MSCs) and then transplanted them into the hind limbs of nude mice to evaluate the potential of angiogenesis. The results showed that the magnetic cell sheet group had more angiogenesis, increased vascular endothelial growth factor expression, and decreased apoptosis. Perea et al. [96] first labeled human smooth muscle cells (SMCs) and human umbilical vein endothelial cells (HUVECs) with MNPs and then used radial magnetic force to drive the cells to efficiently reach the lumen surface of tubular scaffolds, fixed the cells on the matrix surface, and adhered firmly, which effectively promoted the process of vascular endothelialization. To overcome irreversible damage to the endothelial cell layer caused by surgery in repairing blood vessels, resulting in impaired vascular function and restenosis, Vosen and his team [94] combined nanotechnology with gene and cell therapy for site-specific re-endothelialization and restoration of vascular function (Figure 4B). The researchers overexpressed the vascular protection gene endothelial nitric oxide synthase (eNOS) in endothelial cells (ECs) using a complex of lentiviral vectors and MNPs. MNPs-loaded and eNOS-overexpressing cells are magnetic, and even under flow conditions, they can be positioned on the vessel wall in a radially symmetric manner by the magnetic field. The results demonstrated that the treated vessels showed enhanced eNOS expression and activity. Furthermore, the replacement of ECs with eNOS-overexpressing cells restored endothelial function in a mouse model of vascular injury. More interestingly, Mattix et al. [97] added MNPs to the cell spheres through the Janus method and then manipulated the cell sphere to fuse into a vascular tissue structure with a diameter of 5 mm through the magnetic force generated by the external magnetic field (EMF). For the binding of MNPs to materials, Filippi et al. [98] prepared novel magnetic nanocomposite hydrogels by incorporating MNPs into PEG-based hydrogels containing cells from the stromal vascular fraction (SVF) of human adipose tissue; the stimulation of an external static magnetic field (SMF) on the angiogenic properties of the constructs were investigated. The results showed that endothelial cells, pericytes, and perivascular genes were strongly activated, and the expressions of VEGF and CD31(+) were increased. After subcutaneous transplantation in mice, the magnetic drive structure showed denser, more mineralized, and faster-vascularized tissue. Gu et al. [99] studied iron oxide nanoparticles to regulate macrophage phenotype toward M1 polarization and down-regulate M2-related arginase 1 (Arg-1) by affecting the interferon regulatory factor 5 (IRF5) signaling pathway, in which iron-based MNPs are anti-cancer and inhibit tumor angiogenesis, providing new insights. However, MNPs have a concentration-dependent effect on the phenotypic polarization of macrophages. Many studies have shown that low-dose MNPs also can promote M2 polarization, but the related pathway mechanism is rarely studied [100,101]. The aforementioned favorable behaviors based on cell and factor secretion regulation by MNPs can effectively participate in vascular repair.

### 3.3. MNPs as Carriers for Targeted Drug Delivery

MNPs have unique advantages in the construction of drug delivery systems (magnetic drug delivery, MDD), such as inherent magnetic targeting, magnetocaloric drug release, and accessible surface modification, which can maximize drug delivery. By applying a permanent magnet near the target tissue, the accumulation of MNPs at the target site can be induced, reducing the drug’s distribution in the whole body, thereby improving the therapeutic effect and reducing the toxic and side effects [104]. When using MNPs as drug delivery systems, the magnetic properties of nanoparticles are size-dependent, and magnetic nanoparticles with excellent performance can be obtained by adjusting the size. The charge and hydrophobic properties of MNPs affect their interactions with plasma proteins, the immune system, extracellular matrix, or non-targeted cells and determine their biological distribution. Hydrophobic MNPs readily adsorb plasma proteins, leading to recognition by the reticuloendothelial system and eventual clearance from the circulatory system under opsonization, resulting in a short circulating half-life. After modifying its surface with hydrophilic PEG and other molecules, its circulating half-life can be increased. Positively charged MNPs easily bind to non-targeted cells and undergo a nonspecific internalization process. Compared with negatively charged MNPs, positively charged MNPs generally exhibit higher cellular internalization effects [105,106]. In recent years, MDD systems have been widely developed to treat various diseases [107,108], including tumors, such as designing Fe_3_O_4_ nanoparticles-based targeted drug delivery systems to enhance cancer targeting to suppress tumors under static magnetic fields and laser irradiation growth, and the system proved effective for in situ transdermal drug delivery, magnetic fields, and synchronization of laser and biological targeting. Demonstrated in breast cancer models, this system is an effective alternative for the treatment of superficial cancers (Figure 5A) [109]; bone, for example, has developed an exosome derived from neutrophils modified by sub-5 nm ultra-small PBNP (uPB) engineering through click chemistry, which can target deep into cartilage, significantly improve the joint injury of CIA mice, and inhibit the overall severity of arthritis, showing considerable potential in the clinical diagnosis and treatment of arthritis (Figure 5B) [110]; in vascular structures, a developed nanoparticle (MMB-PLGA-PTX) can be used for in-stent restenosis (ISR) treatment that is responsive to external magnetic fields and LIFU. The results showed that magnetic targeting increased the accumulation of MMP-PLGA-PTX 10-fold, while LIFU facilitated the penetration of the released PLGA-PTX into the arterial tissue, thereby increasing the retention time of the released PTX in the stented vascular tissue. Combined with efficacy, this strategy holds great promise for the precise delivery of antiproliferative drugs to stented vascular tissue for ISR therapy (Figure 5C) [111]; skin, such as heme-modified prussian blue nanoparticles (PBNP, an iron-based magnetic nanoparticle) forms a colloid with NO, which is locally dropped at the skin wound site in response to NIR light and releases NO in a targeted and controllable manner to enhance blood Microcirculation, thereby effectively enhancing angiogenesis and collagen deposition during skin wound healing [112] (Figure 5D). In the treatment of vascular injury, the main focus is on treating the etiology [113]. For example, arterial occlusion caused by external force injury or cardiovascular disease can cause severe mortality [114,115], so the rapid recanalization strategy can effectively reduce the risk of death. Intravenous injection of tissue plasminogen activator (tPA) at a fixed dose is the main method to dredge arterial occlusion [116,117]. Still, it will produce complications such as insufficient curative effect and bleeding. Therefore, magnetic drug targeting (MDT) is an effective therapeutic method, which uses an EMF to enhance the specific accumulation of drugs bound to MNPs in the diseased vascular system [118]. Ma et al. [119] first studied the possibility of local thrombolysis with MDT. MNPs combined with tPA (tPA equivalent is 0.2 mg/kg) were used in the rat embolism model. In this study, MNPs administered intravascularly moved and accumulated along the iliac artery affected by thrombus under the action of an external magnet, which resulted in effective targeted thrombolysis and was only less than 20% of the free tPA dose. Atherosclerosis (AS) is also a severe disease that can cause vascular damage [120,121]. Although many drugs can treat atherosclerosis [122,123,124], their systemic administration has serious disadvantages. In particular, the proportion of therapeutic dose reaching atherosclerotic lesions is small, resulting in poor therapeutic effect. Increasing the dose is often impossible in many cases because it can cause serious side effects and drug tolerance. Since the existing treatment strategies for AS are far from ideal, there is an urgent need for targeted therapy as an alternative strategy to exert better therapeutic effects. Cicha et al. [125] developed the combination of dexamethasone on MNPs, which magnetically targeted the balloon injury area in rabbits as well as advanced atherosclerotic plaques. Although the desired effect was not achieved, this may also be due to the selection of candidate drugs. In addition, myocardial infarction caused by coronary plaque rupture can also cause severe inflammation and even heart failure [126,127,128]. Zhang et al. [129] studied in a rat myocardial infarction model, using an in vitro epicardial magnet to accumulate MNPs that bind to the human VEGF gene encoded by an adenovirus vector in the ischemic area. Results showed that targeting MNPs resulted in higher VEGF gene expression in the affected area and better cardiac repair. Currently, the treatment of myocardial infarction with stem cell preparations promises to improve myocardial tissue recovery, but this is still limited due to poor accumulation and retention of therapeutic agents at target sites. Cheng et al. [130] used MNPs to enhance the targeted delivery of cardiac-derived stem cells (CDCs) in female rats with myocardial infarction. Then, a 1.3 T circular magnet was placed about 1 cm above the apex of the heart for 10 min, starting with an intramuscular injection of CDCs. During this process, the naked eye can see slight discoloration of adjacent tissues, suggesting that magnetic particle-labeled CDCs could prevent coronary washout. After 24 h, histology confirmed the retention of magnetic particle-labeled CDCs. Semi-quantitative fluorescence imaging showed that cells spread more in a subgroup of rats injected with non-magnetic or magnetically labeled CDCs without magnets than in rats that received labeled cells and additional magnetically targeted therapy to their lungs and spleen. Subsequently, the SRY gene that was decisively differentiated was analyzed by polymerase chain reaction (PCR). The results showed that CDCs implantation was three times higher in the myocardial tissue of rats in the magnetic target group. Therefore, the authors concluded that magnetic targeting could effectively attenuate the flushing of magnetic-particle-labeled CDCs at the injection site and significantly increase short-term CDCs engraftment in just 10 min. As targeted carriers, MNPs can effectively participate in the treatment of vascular injury. More applications can also be used as magnetic resonance contrast agents for MRI, which can accurately evaluate vascular functional and structural parameters to diagnose and treat.

### 3.4. MNPs as Contrast Agents for Vascular Microenvironment Imaging

Magnetic Resonance Imaging (MRI) is one of the most effective diagnostic imaging tools in medicine, providing clinicians with a high spatial and temporal resolution of biological anatomy and metabolic/functional information in a non-invasive manner. Tissue necrosis, ischemia, and other malignant diseases are of great significance. Under the action of an EMF, different tissues and organs of the organism can generate different resonance signals to form MR images. The strength of the resonance signal is determined by the water content of each part of the body and the relaxation time of water protons. The contrast agent is an image-enhancing contrast agent that can change the body’s relaxation rate of water protons, improve imaging contrast, and display lesions [29,32,131]. MNPs are considered to have promising applications in T2 MRI contrast agents, and especially iron-based MNPs exhibit longer half-lives than clinically used gadolinium-based contrast agents [132] (Figure 6). At present, a variety of iron-based MNPs have been developed as clinical MRI contrast agents for imaging various tissues. For example, the FDA approved Feridex to detect liver lesions, and Combidex has entered the phase III clinical trial stage for the imaging of lymph node metastasis [133]. In terms of vascular structures, in addition to participating in vascular repair in the above ways, MNPs can also be used as vascular microenvironment imaging contrast agents to observe the dynamic changes of vascular graft contour, stenosis or occlusion, and other abnormalities through image visualization to evaluate the process and effect of repair [134]. Flores et al. [135] demonstrated the feasibility of MRI to assess the in vivo performance of tissue-engineered vascular grafts (TEVG) by labeling human aortic smooth muscle cells (HASMCs) with USPIO nanoparticles, which were then seeded into a TEVG and implanted in mice in vivo. The results showed that USPIO-labeled TEVG consistently had sharper boundaries and lowered T2 relaxation time values than unlabeled control scaffolds. In addition, MNPs labeled cells were also used to observe the behavior of related vascular cells by MRI. Perea et al. [136] Labeled HUVECs with clinically approved SPIO, then drove cells to the lumen of polytetrafluoroethylene (PTFE) tubular grafts through a particular electromagnet and then detected endothelial cells with a 1.5 T magnetic resonance scanner to evaluate vascular endothelialization.

### 3.5. Other Role of MNPs in Vascular Repair

MNPs are exposed to alternating EMF, which triggers particle movement and local heating, which produces a high-temperature effect that causes tissue damage in the area around the nanoparticles and have been applied to tumor treatment. The main mechanism of action is to raise the temperature above 42 °C through magnetic heating and lead to protein denaturation, which leads to cell death (Figure 7A). At present, this method has also been an effective means to treat tumor vascular injury. Higher thermal stimulation based on physiological temperature can effectively kill intravascular tumor cells [137,138] and inhibit blood flow to promote the recovery of vascular function [139]. In recent years, studies have also found that MNPs also have biological effects, such as promoting the polarization of macrophages and producing ROS effects (Figure 7B). These effects will also have a significant impact on the vascular repair. Zanganeh et al. [140] found that high concentrations of ferumoxytol can promote macrophage polarization to M1, thereby enhancing the regulation of cancer immunotherapy, including breast cancer, liver cancer, and lung cancer. However, some scholars pointed out that a low concentration of MNPs can also promote the growth of blood vessels [101]. There is no more evidence to prove whether it may regulate the polarization of macrophages at the injured site to M2 type to promote repair. ROS plays important physiological roles in maintaining cardiac and vascular integrity in the cardiovascular system. In particular, it plays a pathophysiological role in cardiovascular dysfunction associated with hypertension, diabetes, atherosclerosis, ischemia-reperfusion injury, ischemic heart disease, congestive heart failure, and violent vascular defects [141,142]. The main ROSs that are important in these processes are superoxide anion (O_2_^−^), hydrogen peroxide (H_2_O_2_), hydroxyl radical (OH), and reactive nitrogen species, among others [143,144]. ROS is both a signaling molecule and an inflammatory mediator, which is central to the progression of many inflammatory diseases [145]. Under normal physiological conditions, ROS of the body is maintained at a low level of dynamic balance and participates in many important physiological processes, such as controlling the inflammatory reaction, killing toxic and harmful substances or tumor cells, promoting leukocyte phagocytosis, responding to growth factor stimulation, participating in the synthesis of biological macromolecules such as prothrombin and collagen, and participating in cell differentiation, proliferation, apoptosis, migration, and other cellular processes [146,147]. For vascular structures, once the body is pathologically damaged, the level of ROS will increase, which can trigger a series of inflammatory reactions, resulting in apoptosis or death of cells, thereby aggravating the damage of vascular structures. An important reason for the high level of ROS at the injury site is that the level of antioxidant enzymes in pathological tissue is lower than that in normal tissue, so it is a strategy to effectively remove ROS and maintain dynamic balance [148,149]. Commonly used antioxidants include vitamins A, C, and E, coenzyme Q10, beta-carotene, superoxide dismutase, and catalase, while commonly used ROS generation inhibitors include NADPH oxidase inhibitors and xanthine oxidase inhibitors [150,151]. The most typical are Prussian blue nanoparticles (PBNPs, a kind of iron-based magnetic nanoparticle), which are nanozymes just discovered in recent years, which have incomparable advantages to other nanozymes, such as multi-enzyme activity, high catalytic rate, excellent ROS scavenging ability, and good biological safety. Nowadays, PBNPs have received extensive attention in treating inflammatory diseases and show great potential for application [152,153,154,155,156]. Zhang et al. [70] utilized the scavenging properties of hollow Prussian blue nanoparticles (HPBZs) to treat ischemic stroke in rats. HPBZs can not only reduce oxidative stress but also inhibit apoptosis, alleviate inflammation, and improve the tolerance of ischemic brain injury. Zhao et al. [71,157] Used PBNPs and their analogs as active ingredients to treat inflammatory bowel disease. PBNPs can not only clear ROS but also effectively reduce inflammatory factors and have a good therapeutic effect on intestinal inflammation induced by dextran sodium sulfate. In conclusion, these results may be candidates for vascular repair applications, especially for vascular defects.

MNPs participate in vascular injury repair based on their unique physicochemical properties. However, no matter how it participates in vascular repair, MNPs will pass through the blood circulatory system, affecting the vascular wall’s function, blood pressure, or hemodynamics. The most typical is that when INOPs are less than 7 nm, they will leak out of the vascular structures and be discharged by the kidney, while 200 nm–4 μM particles are easily phagocytized by macrophages of the mononuclear phagocytosis system (MPS). Therefore, the development of INOPs for vascular usually needs to be at 10–200 nm [158,159]. Secondly, we know that the endothelial cell layer is the innermost layer of the vascular wall, which can maintain the hemostasis and smooth blood flow of vascular structures by releasing NO, heparin, plasmin, and other regulatory molecules [160,161]. The instability of INOPs sometimes leads to the release of iron ions, resulting in the dysfunction of most organelles in endothelial cells, such as lysosome, golgi apparatus, endoplasmic reticulum, and mitochondria, which in turn induces oxidative stress, inflammation, and gene mutation, and finally leads to the destruction of the endothelial cell layer, the impairment of vascular wall function, and thrombosis [162]. In addition, usually naked INOPs are prone to aggregate in complex saline solutions (such as blood), adversely affecting living tissue or occluding vascular structures. Stable anti-aggregation coatings, such as serum albumin, can greatly improve IONP dispersibility [163,164,165]. More importantly, studies have shown that the concentration of ions also significantly affects blood pressure and hemodynamics. For example, in atherosclerotic coronary arteries, the deposition of fibrofatty plaque reduces the elasticity of the arterial wall and may ultimately increase pulse pressure. This can lead to advanced disease within the large arteries [166]. Injecting a suspension of nanoparticles as a drug carrier into the bloodstream changes the viscosity of the blood and causes a pressure drop in atherosclerotic coronary arteries. However, these are related to the concentration of nanoparticles used, and high concentrations of nanoparticles may also increase blood pressure and even reduce hemodynamic effects [167]. Therefore, a suitable nanoparticle concentration is also crucial. In conclusion, the repair of vascular structures based on MNPs shows great potential for clinical application, but its impact on the vascular microenvironment cannot be ignored.

**Figure 7 pharmaceutics-14-01433-f007:**
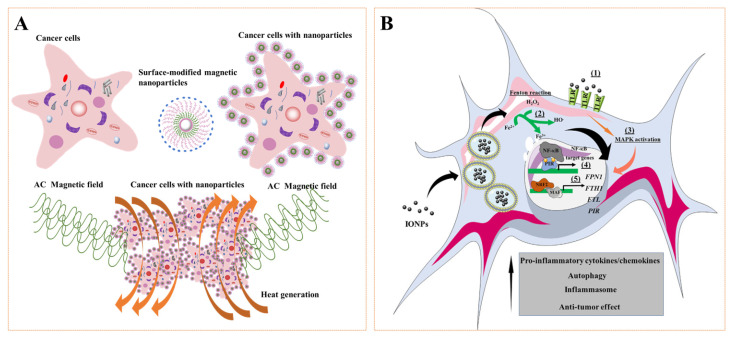
(**A**) General procedure for magnetic hyperthermia of tumor cells. AC: alternating current. (**B**) Overview of the effects of iron oxide nanoparticles (IONPs) on macrophage polarization. Reprinted with permission from Ref. [168]. Copyright 2017 Copyright Mulens-Arias V. et al.

## 4. Conclusions and Outlook

Vascular injury caused by various reasons seriously affects human health, which is also one of the significant challenges clinical surgery faces. MNPs play important roles in diagnosing, treating, and repairing vascular structures based on their unique magnetic properties, including magnetic stimulation, magnetic actuation, magnetic resonance imaging, magnetic hyperthermia, magnetic-assembled scaffolds, and magnetically targeted drug delivery. The occurrence of these magnetic response behaviors can significantly affect the scaffold performance, cell behavior, factor secretion, and drug delivery related to vascular repair, thereby effectively promoting the rapid repair of vascular structures. However, there are still challenges in the large-scale clinical application of MNPs, which can only be reflected in the experimental academic environment, and the challenges in the human body will be more prominent. Under the fact that the physiological environment of human vascular cannot be changed, the preparation of high-performance MNPs suitable for vascular repair is the key to clinical application. For example, MNPs must be small enough to move freely between various vascular structures when dispersed in blood to avoid vascular embolism. Studies have also found that MNPs at the 10–100 nm scale could maintain longer blood circulation time; high saturation magnetization, the strong driving force at low doses in drug-targeted treatment of vascular injury, and a high temperature rise effect at low doses in magnetic hyperthermia repair of vascular structures; and functional coating in the vascular physiological environment, in which MNPs will be adsorbed on the surface by plasma proteins in the blood and proteins and coagulation factors in the complement system to form the so-called blood protein corona, covering the original surface-design nanoparticles (NPs) and disturbing the recognition receptors of target cells. Therefore, polyethylene glycol is usually coated to reduce corona formation so that they can be recognized by immune cells quickly and clearly. Biological effects, magnetically assembled scaffolds capable of rapid re-endothelialization, are crucial for rapid re-endothelialization the repair of vascular rupture damage. The loaded MNPs should have the ability to regulate the microenvironment of the injury site to promote rapid reendothelialization, such as affecting the migration and polarization of immune cells, including macrophages and neutrophils. However, the problems to be solved by high-performance MNPs suitable for vascular repair are not limited to the above description but include crystallinity, uniformity, surface properties, shape, hydrophilicity, and hydrophobicity.

## Figures and Tables

**Figure 1 pharmaceutics-14-01433-f001:**
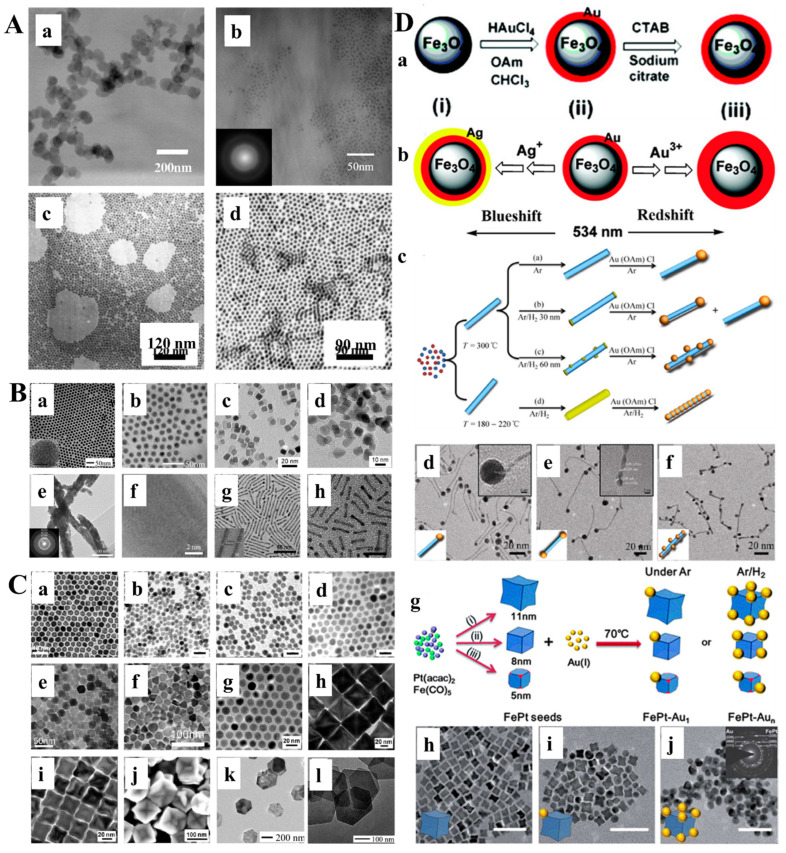
(**A**) Single-component metal nanoparticles. (**a**) TEM image of 65 nm Fe NPs; (**b**) TEM image of 3.7 nm Ni NPs; (**c**,**d**) TEM images of (**c**) 6 nm and (**d**) 9 nm Co NPs. (**B**) Metal alloy nanoparticles. (**a**,**b**) TEM images of (**a**) 11 nm and (**b**) 16 nm FePd NPs; (**c**,**d**) TEM images of (**c**) fcc-FePt@Fe_3_O_4_ and (**d**) fct-FePt NPs; (**e**) TEM image of FePt NWs and (**f**) HRTEM image after annealing; (**g**,**h**) TEM images of (**g**) FePt NWs and (**h**) FePt NRs. (**C**) Metal oxide nanoparticles. (**a**) TEM image of 16 nm Fe_3_O_4_ NPs; (**b**–**d**) TEM images of (**b**) 7 ± 0.5 nm, (**c**) 8 ± 0.4 nm, and (**d**) 10 ± 0.8 nm Fe_3_O_4_ NPs. The scale bars are 20 nm; (**e**) TEM image of octahedral Fe_3_O_4_ NPs; (**f**) TEM image of Fe_3_O_4_ nanoprisms; (**g**–**i**) TEM images of (**g**) 14 nm spherical and (**h**) 32 nm and (**i**) 53 nm truncated octahedral FeO NPs and (**j**) SEM image of truncated octahedral FeO NPs; (**k**,**l**): TEM images of (**k**) Co(OH)_2_ and (**l**) CoO NPLs. (**D**) Heterostructures. (**a**,**b**) Schematic illustrations of the syntheses of (**a**) Fe_3_O_4_@Au [(**i**) Fe_3_O_4_; (**ii**) Fe_3_O_4_@Au; (**iii**) hydrophilic Fe_3_O_4_@Au] and (**b**) Fe_3_O_4_@Au@Ag; (**c**) Schematic illustration of FePt−Au HNWs. FePt NWs were first prepared at high temperature, and then Au nanocrystals were grown onto them to form FePt−Au HNWs. By control over the amount of H2, different morphologies were obtained, such as (**d**) tadpole-like, (**e**) dumbbell-like, and (**f**) bead-like HNWs; (**g**) Schematic illustration of FePt−Au heterostructured NPs and (**h**−**j**) TEM images of (**h**) FePt concave nanocubes and (**i**) FePt−Au1 and (**j**) FePt−Aun heterostructured NPs. The scale bars are 50 nm. Reprinted with permission from Ref. [44]. Copyright 2018 Copyright Zhu Y. et al.

**Figure 2 pharmaceutics-14-01433-f002:**
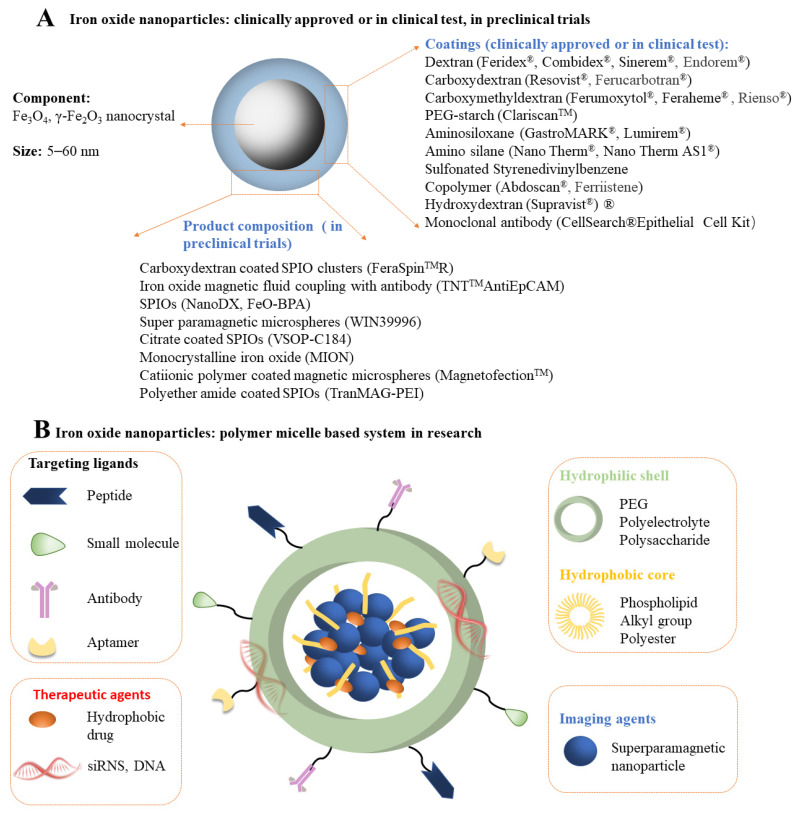
(**A**) Iron oxide nanoparticles (IONPs) approved or in (pre-) clinical trials [27,31]; the coating material is listed on the left, and the commercial product name is shown in the brackets on the right. SPIOs: super paramagnetic iron oxides. (**B**) IONPs: polymer micelle-based system in research.

**Figure 3 pharmaceutics-14-01433-f003:**
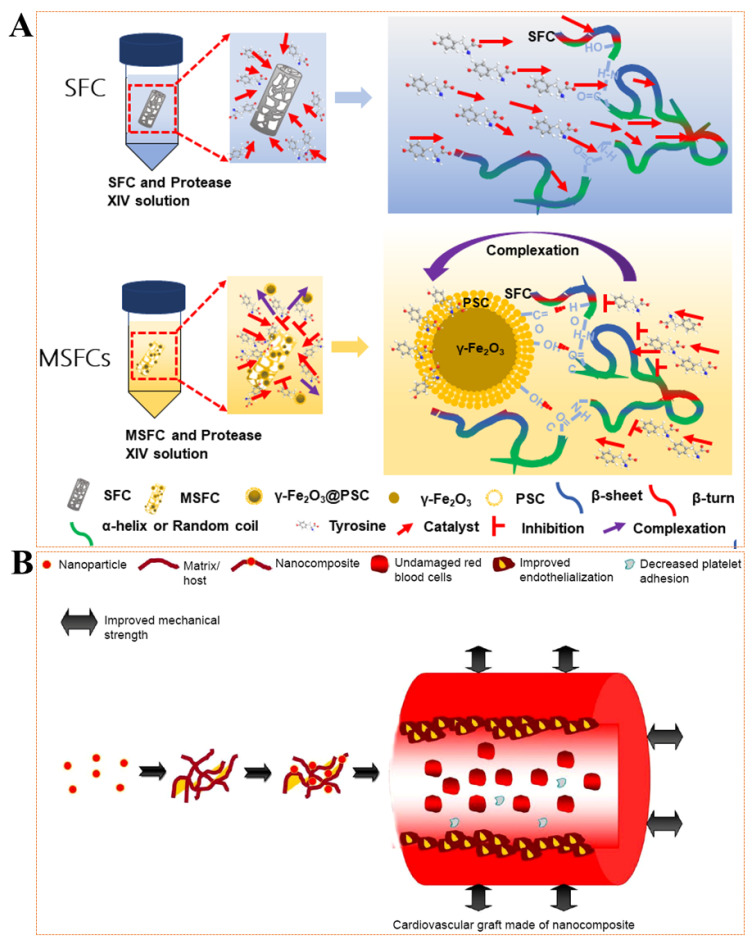
(**A**) Significantly delayed degradation of silk fibroin scaffolds (SFC) loaded with MNPs. Reprinted with permission from Ref. [16]. (**B**) Using nanocomposites to improve the performance of cardiovascular grafts. Reprinted with permission from Ref. [91]. Copyright 2015 Copyright Jaganathan S.K. et al.

**Figure 4 pharmaceutics-14-01433-f004:**
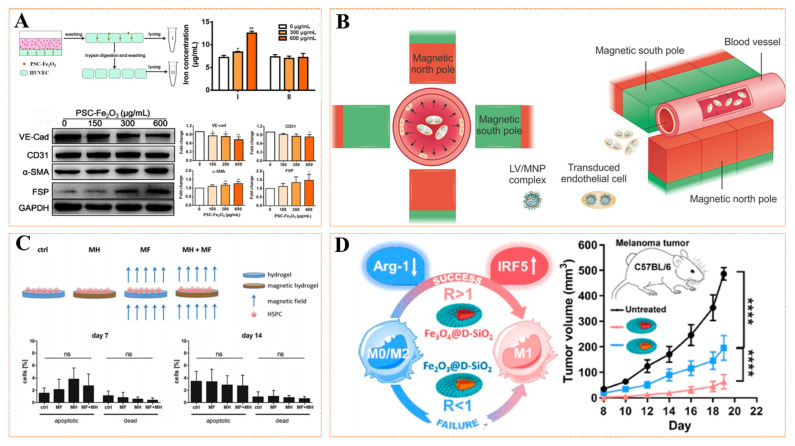
(**A**) Magnetic nanoparticles (Fe_2_O_3_@PSC)-labeled VECs and their effects on the expression of related factors. Reprinted with permission from Ref. [102]. Copyright 2019 Copyright Wen T. et al. (**B**) The eNOS, a vascular protective gene, was overexpressed in ECs using a complex of lentivirus vectors and MNPs, and then localized on the vessel wall in a radially symmetric manner by a magnetic field. Reprinted with permission from Ref. [94]. Copyright 2016 Copyright Vosen S. et al. (**C**) Effects of magnetic macroporous hydrogels prepared by magnetic nanoparticles composite PEG hydrogels on the function of human hematopoietic stem and progenitor cells (HSPCs). Reprinted with permission from Ref. [103]. Copyright 2018 Copyright Rodling L. et al. (**D**) Magnetic nanoparticles (Fe_3_O_4_@D-SiO_2_ and Fe_2_O_3_@D-SiO_2_) rely on the IRF-5 signaling pathway to polarize M1 and downregulate M2-related Arg-1, thereby affecting the expression of related cytokines, and can be used for tumor vascular injury therapy. Reprinted with permission from Ref. [99]. Copyright 2019 Copyright Gu Z. et al. The results were considered significant at * *p* < 0.05, ** *p* < 0.01, *** *p* < 0.001, **** *p* < 0.0001.

**Figure 5 pharmaceutics-14-01433-f005:**
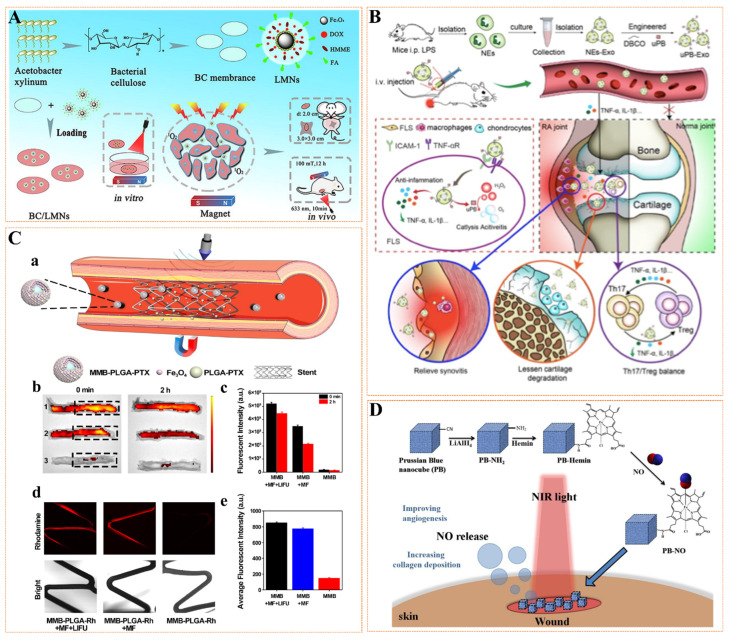
(**A**) The laser-sensitized magnetic nanoparticles (LMN) were constructed using Fe_3_O_4_ nanoparticles as the core, coated with doxorubicin and hematoporphyrin mether hydrogel, and then coated with dendritic acid on the surface of the composite. LMN was then loaded into the bacterial cellulose (BC) membrane. Synthetic BC/LMN enhances transdermal drug targeting for breast cancer when administered via magnetic field and laser. Reprinted with permission from Ref. [109]. Copyright 2019 Copyright Zhang L.K. et al. (**B**) Surface-engineered neutrophil-derived exosomes (NEs-Exo) and ultra-small Prussian blue magnetic nanoparticles (uPB) effectively treat advanced rheumatoid arthritis by clicking on the chemically mediated repair of the inflammatory environment. Reprinted with permission from Ref. [110]. Copyright 2022 Copyright Zhang L. et al. (**C**) Schematic diagram of magnetic guided and ultrasonic stimulation for the treatment of ISR with targeted and deep penetrating delivery strategies (**a**); Evaluation of MMB-PLGA Rhdomain targeting and penetration of isolated porcine coronary arteries. (**b**) Representative fluorescence images of the stented porcine coronary arteries at 0 min and 2 h after different treatments; (**c**) Quantification of the fluorescence intensity in the areas within the black dotted frames at 0 min and 2 h post treatments; (**d**) Representative fluorescence and bright field images of the stents at 2 h post treatments; (**e**) Quantification of the fluorescence intensity of the stents shown in the fluorescence images. Reprinted with permission from Ref. [111]. Copyright 2020 Copyright Wang S. et al. (**D**) Preparation of NO-carried Prussian blue (PB-NO) nanocubes for the treatment of incisional wounds. Reprinted with permission from Ref. [112]. Copyright 2019 Copyright Su C.H. et al.

**Figure 6 pharmaceutics-14-01433-f006:**
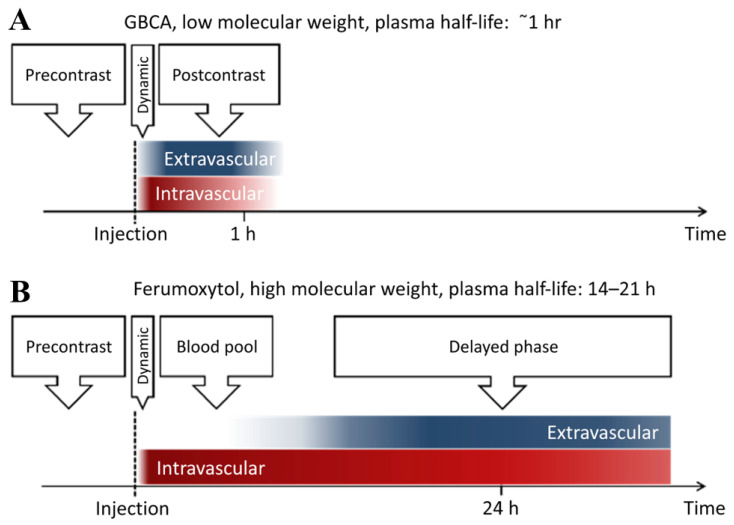
(**A**) Gadolinium-based contrast agent (GBCA) and (**B**) ferumoxytol (a superparamagnetic iron oxide nanoparticle)-reinforced phase. Compared with GBCA, ferumoxytol shows a longer intravascular half-life. Reprinted with permission from Ref. [132]. Copyright 2017 Copyright Toth G.B. et al.

## Data Availability

Not applicable.

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
