# Peer review of "Vascular Repair by Grafting Based on Magnetic Nanoparticles"

_pharmaceutics, 2022, doi:10.3390/pharmaceutics14071433_

Round 1

Reviewer 1 Report

The topic of the manuscript is actual, the manuscript is well written and structured. It focuses mainly on the influence of iron NPs on blood vessels - on their use in the treatment and repair of blood vessels, which is important for future use in medical applications.

The only "problem" with reading of this manuscript arises from the large number of abbreviations, which impair the comprehension and fluent reading of the text. The manuscript would benefit if the number of abbreviations was reduced wherever possible and a List of abbreviations would be included.

The authors could also consider inserting a short paragraph dealing with the effect of IONs on vascular wall function (and thus on changes in blood pressure), as this could be one of the limiting factors in the use of IONs in humans.

Author Response

Reviewer #1: The topic of the manuscript is actual, the manuscript is well written and structured. It focuses mainly on the influence of iron NPs on blood vessels - on their use in the treatment and repair of blood vessels, which is important for future use in medical applications.

Reply: We thank the referee for the positive comments and appreciation of this work.

Comment 1: The only "problem" with reading of this manuscript arises from the large number of abbreviations, which impair the comprehension and fluent reading of the text. The manuscript would benefit if the number of abbreviations was reduced wherever possible and a List of abbreviations would be included.

Reply: Thank you very much for your constructive comments, which can increase the readability of our review. According to the reviewers' suggestions, we have added the whole list of abbreviations in the supplementary information of the revised manuscript (See Supplementary information).

Comment 2: The authors could also consider inserting a short paragraph dealing with the effect of IONs on vascular wall function (and thus on changes in blood pressure), as this could be one of the limiting factors in the use of IONs in humans.

Reply: Thank you very much for your positive suggestions, which can greatly enrich the content of the article. We have supplemented relevant contents in the revised manuscript (Page 14, line 449).

Reviewer 2 Report

The work Vascular repair based on magnetic nanoparticles designed by Xin Liu and co-authors is a brief overview of the latest trends in the application of magnetic nanoparticles in a small but vital area: vascular injury. The manuscript is well structured, the quality of the text is good, and each statement is explicitly argued. However, some minor issues need to be considered for the manuscript to be published.

1.       Add in the introduction some of the reviewers that have been made in the field of vascular injury and applications of MNP in this field, showing the need to publish other essays on this topic, motivating the need for this review.

2.       Some references, such as 25 and 26, are far too general and do not adequately reflect the subject under discussion.

3.       The third figure on page 7 is incorrectly numbered as Figure 2. Change Figure 2 to Figure 3 on page 7 and renumber the other figures accordingly.

4.       In Figure 2 on page 7 (corrected 3), the images in 2B should become 2A, because they are the first discussed in the text, and A should become B.

5.       Figures are rarely cited in the text, so sometimes the text refers to certain results in the figures, but does not refer to the figure. An example is the phrase between lines 172-174 (at the end of which refers to Figure 2B) should have been found (corrected variant 3B); phrase 186-189 with references to Figure 2B (corrected variant 3B); the phrase between lines 220-223 should refer to Figure 4B (corrected 5B) and so on. In my opinion, the figures should be noted in the text as explicitly as possible so that the reader can more easily understand the arguments.

6.       Some of the results in the figures should be discussed in more detail in the text body. For example, Figure 3A on page 9 contains relevant results that are not discussed at all in the text. Please check if other information is poorly highlighted, in order to better shows the results published in the field. Another example, Figure 4 is not quoted/discussed in the text, the results there are not explained, although it contains very interesting information regarding MNPs as magnetically guided delivery agents.

7.       Chapter, “5. Other role…”, please add some paragraphs to show the role of the inflammation process and reactive oxygen species in vascular injury and add some recent results regarding magnetic nanoparticles with an anti-inflammatory and antioxidant effect which may be candidates for vascular repair applications.

In my opinion, this topic is very current and of overwhelming importance, so the publication of a review on this topic is welcome. The authors have raised the issue, but the draft cannot be published without the proposed corrections, which, in my opinion, are minor and more related to the organization of the material.

Author Response

Reviewer #2: The work Vascular repair based on magnetic nanoparticles designed by Xin Liu and co-authors is a brief overview of the latest trends in the application of magnetic nanoparticles in a small but vital area: vascular injury. The manuscript is well structured, the quality of the text is good, and each statement is explicitly argued. However, some minor issues need to be considered for the manuscript to be published.

Reply: We thank the referee for the positive comments and appreciation of this work!

Comment 1: Add in the introduction some of the reviewers that have been made in the field of vascular injury and applications of MNP in this field, showing the need to publish other essays on this topic, motivating the need for this review.

Reply: Many thanks to the reviewer for their professional and valuable comments on our work, which are very helpful to improve the quality and readability of the article. We have also supplemented this section in the "Introduction" section based on the reviewer's comments (Page 2, line 51).

Comment 2: Some references, such as 25 and 26, are far too general and do not adequately reflect the subject under discussion.

Reply: We thank the referee for pointing this out. Based on reviewer suggestions, we have corrected 25,26 papers to bring them closer to the topic of discussion, replacing it with [25], [26] .

Comment 3: The third figure on page 7 is incorrectly numbered as Figure 2. Change Figure 2 to Figure 3 on page 7 and renumber the other figures accordingly.

Reply: Corrected, thanks.

Comment 4: In Figure 2 on page 7 (corrected 3), the images in 2B should become 2A, because they are the first discussed in the text, and A should become B.

Reply: Thanks to the reviewers for their valuable comments. According to the reviewers' suggestions, we have changed the order of the Figure. See Figure 3 for details.

Comment 5: Figures are rarely cited in the text, so sometimes the text refers to certain results in the figures, but does not refer to the figure. An example is the phrase between lines 172-174 (at the end of which refers to Figure 2B) should have been found (corrected variant 3B); phrase 186-189 with references to Figure 2B (corrected variant 3B); the phrase between lines 220-223 should refer to Figure 4B (corrected 5B) and so on. In my opinion, the figures should be noted in the text as explicitly as possible so that the reader can more easily understand the arguments.

Reply: Many thanks to the reviewers for their excellent comments, which are very important to increase the reader's understanding. We have made changes according to the reviewer's suggestions, please see the revised manuscript for details (Page 6, line173 and 187; Page 8, line 222).

Comment 6: Some of the results in the figures should be discussed in more detail in the text body. For example, Figure 3A on page 9 contains relevant results that are not discussed at all in the text. Please check if other information is poorly highlighted, in order to better shows the results published in the field. Another example, Figure 4 is not quoted/discussed in the text, the results there are not explained, although it contains very interesting information regarding MNPs as magnetically guided delivery agents.

Reply: Many thanks to the reviewers for their comments. We have reorganized the article and believe that the questions raised by the reviewers are very reasonable. I would also like to kindly explain to the reviewers that there is no relevant discussion on the illustrations, mainly because I want to make a supplementary explanation based on the viewpoints in front of the illustrations so that readers can better understand (For example, the original figure 4). In fact, starting from Lshii et al. [98], we present a detailed discussion of the above points to avoid repetition. In fact, a relatively brief description is also given below the legend, and readers can see it at a glance. However, the reviewers' suggestions are excellent, and we have supplemented the entire article with figures that need to be discussed in revised manuscript. Once again, we thank the reviewers for their professional and helpful suggestions (Page 9, line 278).

Comment 7: Chapter, “5. Other role…”, please add some paragraphs to show the role of the inflammation process and reactive oxygen species in vascular injury and add some recent results regarding magnetic nanoparticles with an anti-inflammatory and antioxidant effect which may be candidates for vascular repair applications.

Repy: The questions raised by the reviewers are very good. I'm sorry we missed an important content. According to the reviewers' suggestions, we have supplemented this part in the revised manuscript (Page 13, line 413).

Reviewer 3 Report

The paper is very difficult to read because it is not structured and all information is given at the same level of importance.

First of all, the title should account for the fact that MNPs are used in graft and not generally for vascular repair, then it should be changed in ' Vascular repair by grafting based on MNP' or something similar.

The overview of the MNPs should be separated into sub-chapters on composition, preparation and imaging properties.

MNPs in the vascular graft should be separated into sub-chapters on principles and applications, targeted drug delivery, imaging and other applications.

By applying such a structure it will be easier to avoid repetitions and duplications, that are now present and are quite disturbing for the reader

Author Response

Reviewer #3: The paper is very difficult to read because it is not structured and all information is given at the same level of importance.

Reply: We sincerely apologize to the reviewer for the article is difficult to read. The team and I read the article carefully and found some problems with the subtitle. We are very sorry for this and have revised it in the revised manuscript.

Comment 1: First of all, the title should account for the fact that MNPs are used in graft and not generally for vascular repair, then it should be changed in ' Vascular repair by grafting based on MNP' or something similar.

Reply: The reviewers raised a good question, and we thought about it as we wrote. Since MNPs as a treatment for the cause of vascular injury was written in “3.3 MNPs as carriers for targeted drug delivery”, the title of the original manuscript was written at that time. After our discussion, it was felt that the reviewer's title was more focused, so we are happy to adopt the title (Vascular repair by grafting based on MNP)suggested by the reviewers. Once again, we are very grateful to the reviewers for their professional comments (See the title and subtitle of the revised manuscript).

Comment 2: The overview of the MNPs should be separated into sub-chapters on composition, preparation and imaging properties.

Reply: Thank you for the careful review. Based on reviewer comments, we have divided "2. Overview of Magnetic nanoparticles (MNPs)" into sub-chapters in the revised manuscript (See revised manuscript,2.1; 2.2).

Comment 3: MNPs in the vascular graft should be separated into sub-chapters on principles and applications, targeted drug delivery, imaging and other applications.

Reply: Thanks to the reviewers for their careful reading of the article. The team and I read the article carefully and found some problems in the subtitle, which may be due to negligence in writing. In fact, we searched previous manuscripts, which were divided into chapters as suggested by the reviewers. We are very sorry for this and have revised it in the revised manuscript (See revised manuscript, 3.1; 3.2; 3.3; 3.4; 3.5)

Comment 4: By applying such a structure it will be easier to avoid repetitions and duplications, that are now present and are quite disturbing for the reader.

Reply: Thank you very much for your questions about the structure. In fact, we carefully searched the original version and found that its structure was divided into chapters according to the reviewer's suggestions. Perhaps due to the mistakes in editing the article, the reviewer's version was different. We apologize for the confusion caused to the reviewer. At present, we have revised it in the revised manuscript. Thanks again to the reviewers (See revised manuscript).

Round 2

Reviewer 3 Report

The paper is now better organized and readable, but it still requires an extensive linguistic revision to avoid repetitions of the same words in the sentence and a more effective explanation of what the authors intend to say.